# Evaluation of Stress Distribution during Insertion of Tapered Dental Implants in Various Osteotomy Techniques: Three-Dimensional Finite Element Study

**DOI:** 10.3390/ma14247547

**Published:** 2021-12-09

**Authors:** Jaideep Mahendra, Yemineni Bhavan Chand, Little Mahendra, Hytham N. Fageeh, Hammam Ibrahim Fageeh, Wael Ibraheem, Khaled M. Alzahrani, Nasser M. Alqahtani, Nasser Mesfer Alahmari, Mohammad Almagbol, Ali Robaian, Nasina Jigeesh, Saranya Varadarajan, Thodur Madapusi Balaji, Shankargouda Patil

**Affiliations:** 1Department of Periodontics, Meenakshi Ammal Dental College and Hospital, Chennai 600095, India; 2Department of Dental and Oral Surgery, ASRAM Medical College, Hospital & Research Centre, West Godavari District, Eluru 534005, India; bhavanchand@gmail.com; 3Department of Periodontics, Maktoum Bin Hamdan Dental University, Dubai 213620, United Arab Emirates; littlemahendra24@gmail.com; 4Department of Preventive Dental Sciences, College of Dentistry, Jazan University, Jazan 45142, Saudi Arabia; hfageeh@jazanu.edu.sa (H.N.F.); hafageeh@jazanu.edu.sa (H.I.F.); Wibraheem@jazanu.edu.sa (W.I.); 5Department of Prosthetic Dental Sciences, College of Dentistry, Prince Sattam Bin AbdulAziz University, Alkharj 11942, Saudi Arabia; dr_kmq@hotmail.com; 6Department of Prosthodontics, College of Dentistry, King Khalid University, Abha 61421, Saudi Arabia; nmalqahtani@kku.edu.sa (N.M.A.); nmalahmari@kku.edu.sa (N.M.A.); 7Department of Community and Periodontics, Faculty of Dentistry, King Khalid University, Abha 61421, Saudi Arabia; malmagbol@kku.edu.sa; 8Conservative Dental Sciences Department, College of Dentistry, Prince Sattam bin Abdulaziz University, Alkharj 11942, Saudi Arabia; Alkharjali.alQahtani@psau.edu.sa; 9Department of Operations & Information Technology, IBS Hyderabad, ICFAI Foundation for Higher Education, Hyderabad 501203, India; jigeeshn@gmail.com; 10Department of Oral Pathology and Microbiology, Sri Venkateswara Dental College and Hospital, Chennai 600130, India; vsaranya87@gmail.com; 11Tagore Dental College and Hospital, Chennai 600127, India; tmbala81@gmail.com; 12Department of Maxillofacial Surgery and Diagnostic Sciences, Division of Oral Pathology, College of Dentistry, Jazan University, Jazan 45412, Saudi Arabia

**Keywords:** finite element analysis, von mises, osteotomy, bone tap, countersink, cortical bone, stress distribution, implant

## Abstract

Conventional osteotomy techniques can, in some cases, induce higher stress on bone during implant insertion as a result of higher torque. The aim of the present study was to evaluate and compare the stress exerted on the underlying osseous tissues during the insertion of a tapered implant using different osteotomy techniques through a dynamic finite element analysis which has been widely applied to study biomedical problems through computer-aided software. In three different types of osteotomy techniques, namely conventional (B1), bone tap (B2), and countersink (B3), five models and implants designed per technique were prepared, implant insertion was simulated, and stress exerted by the implant during each was evaluated. Comparison of stress scores on the cortical and cancellous bone at different time points and time intervals from initiation of insertion to the final placement of the implant was made. There was a highly statistically significant difference between B1 and B2 (*p* = 0.0001) and B2 and B3 (*p* = 0.0001) groups. In contrast, there was no statistically significant difference in the stress scores between B1 and B3 (*p* = 0.3080) groups at all time points of implant placement. Overall, a highly significant difference was observed between the stresses exerted in each technique. Within the limitations of our study, bone tap significantly exerted lesser stresses on the entire bone than conventional and countersink type of osteotomy procedures. Considering the stress distribution at the crestal region, the countersink showed lower values in comparison to others.

## 1. Introduction

Dental implants represent an important advancement in oral rehabilitation. They have been widely accepted as standard protocol for replacing missing teeth. Implants have shown a success rate of approximately 95% for a period of 16 years in treating fully or partially edentulous patients compared to the other traditional treatment modalities [1]. The salient feature of dental implants is their biocompatibility when placed in the bone. Their prime function is to take up and transfer the occlusal load to the supporting underlying bone of the dental implant [2]. The long-term stability of an implant is a function of both procedural (osteotomy technique, design of the implant, prosthetic materials, etc.) and patient-related factors (bone quality and/or bone volume). Control over these factors is vital for the success and stability of an implant. Increasing contact between the bone and the implant results in higher forces exerted on the surrounding bone. Three types of forces act on the implant: compressive, tensile, and shear. Compressive loads tend to maintain the implant-bone interface. Tensile and shear forces tend to disrupt this interface. Shear forces can be destructive to the bone and implant [3].

The preparation of implant sites impacts the ultimate end bone-implant contact. The mode of preparation reflects the stress generated by the implant on the surrounding bone [4]. Conventional cylindrical osteotomy preparation is done in routine dental practice. Over time, these conventional techniques have undergone modification. Countersink and bone tapping before the insertion of the implant is thought to reduce the high stress and control the higher torque values.

After osteotomy, the implant is inserted manually or with the aid of a motor. During insertion, the implant compresses and penetrates the bone by exerting stress on the surrounding bone. Bone is a pliable tissue and reacts well to the changes in temperature and stress acting on it [5]. After osteotomy, an area of devitalized bone is seen around the bony cavity, which is further rejuvenated by the fresh blood supply around the site [6]. The adjacent bone reacts to the stress exerted by the implant through complex biomechanical processes that may be detrimental to osseointegration. The magnitude of stress generated during the insertion is directly proportional to the bone damage and healing. Less stress on the bone allows smaller micromotions between the implant and the bone and enables optimal implant osseointegration [4]. However, a high insertion torque is sometimes required in the case of immediate loading. Thus, applying a higher magnitude of stress in osteotomy preparation also depends upon the treatment protocol.

High stress could result in fibro integration and lead to an early implant failure [7].

The method of osteotomy chosen can influence bone loss. The stress acting on the bone around the implant determines the extent of resorption. Earlier studies have suggested that the type of surgical preparation and stress acting on bone during implant placement is directly proportional to bone loss [8,9,10]. According to Wolff’s theory, the bone’s response to resorption is directly proportional to the stress generated within the bone [11,12]. The literature reveals that the surgical preparation technique influences the forces during implant insertion and impacts the biomechanical, clinical, and biological effects seen after implant insertion [13,14,15,16].

Finite element analysis (FEA) is a computational technique used to analyze and evaluate the stress and deformation on implants, implant components, and bone by discretization [13,14]. FEM can serve to analyze complex tissues when in vivo tests are not feasible. It can determine the behavior of a structure subjected to a certain load through a mathematical model. Previous studies have successfully used the finite element method (FEM) in examining dental implants [17,18,19,20].

Several studies have reported the relationship between the implant geometry and force distribution in a static condition [2,19,21,22]. There is a need to evaluate the force distribution within the surrounding bone during the dynamic insertion of the implant. The osseous preparation technique selected for dental implant placement can influence the variations in the stress distribution. This study aims to compare the stress exerted during implant insertion using three different osteotomy techniques through FEM analysis.

## 2. Materials and Methods

Tapered dental implant and the mandible were designed and modeled in the ANSYS FEA system (Ansys Softwares, Canonsburg, PA, USA). Bone dimensions were obtained from computed tomography images of a human jawbone.

### 2.1. Preparation of Models

Standard tapered implant dimensions of 4.5 mm diameter and 11 mm length were considered for specification and standardization [20]. Fifteen cut sections of the mandibular model were prepared for the study [23]. The models were made and auto-meshed with tetrahedral elements. The simulated bone model is composed of the outer cortical bone, and inner cancellous bone. The cancellous and cortical bones were assumed to be orthotropic/isotropic, homogenous, linearly elastic, and the prepared site was presumed to be without defects.

The bone sections were divided into three groups of 5 models based on the different types of osteotomy techniques. All the bone sections had similar bone physical properties designed from the procured same human mandibular cadaveric model. The three test groups were modeled within the bone and named B1, B2, and B3, which resembled the shape of the final drill. A bone cavity with parallel walls was prepared without any tapering (Figure 1).

B1 group (control): A conventional bone cavity made with 4.25 mm diameter and 11 mm length of implant dimensions. Conventional osteotomy was considered the control group. The conventional bone cavity consisted of no modification where neither the cortical crest region nor the bone tap-inducing screw threads were performed.

B2 group (Bone tap): Bone cavity prepared by modeling threads respective to the implant being inserted.

B3 group (Countersink): Bone cavity prepared with a 1.2 mm deep and 0.5 mm excess diameter preparation than B1 on the crest. Countersink osteotomy preparation was done in this group. The preparation was done at the crestal region, apically towards the bone cavity.

### 2.2. Structure of FEM

The number of elements and nodes in the FEM was 60,193 and 10,253, respectively. A convergence test of 10% determined the number of control elements of the mesh of 370.345. The contact points within the FEM were 95. The biomechanical behavior of cortical and cancellous bone was simulated through Young’s modulus, Poisson’s ratio, density, yield stress, and plastic strain values [24]. The friction coefficient between bone and commercially pure titanium was obtained using fretting wear tests performed in a salt solution. The point of fracture determined the stress–strain relationship of the bone (Table 1).

### 2.3. Simulation of the Implant Insertion

Dental implants were inserted into the bone cavity with a 90-degree mandibular segment to compensate for the computational requirements [26]. Dynamic insertion of the implant was done with a displacement of 0.2 mm/s and angular displacement of 2.1 radian/s. A constant vertical linear velocity was calculated to the number of pitches. The total insertion period of the fixture was 57 s. The tangential properties at the bone-implant interface were defined using a friction coefficient, and the standard features were established as hard contact. A simplified model was adopted as a sophisticated model of the implant would create displacement of bone elements, disengaging the fragments.

### 2.4. Stress Measurement

Stress patterns and force distributions on the surrounding osseous tissues were evaluated. The von Mises stress in both cortical and cancellous bone was recorded along a predetermined line on the vertical axis, beginning with the implant’s initial contact with the bone until the implantation process was completed. The highest von Mises stress value and pattern of the stress distribution were compared within B1, B2, and B3 under similar insertion conditions. Stress distribution was recorded at intervals of 5 s from 0–57 s.

### 2.5. Statistical Analysis

The three groups were compared in terms of stress distributions on the mandibular cut section models. Mean ± SD was used to compare stress scores at different time points for the three groups. The data was tabulated in MS Excel, and statistical analysis was done using SPSS v17.0. One-way ANOVA was used for comparison between the groups. The *p*-value was calculated using the Kruskal–Wallis test ANOVA method. Mann–Whitney U test for used for pair-wise comparison of the stress scores. *p* < 0.05 was considered significant.

## 3. Results

### 3.1. Comparison of Stress Scores in the Cancellous Bone between B1, B2, and B3 Groups at Different Time Points

The stress scores between B1-B2 (*p =* 0.0001) and B2-B3 (*p =* 0.0001) groups were highly statistically significant. Stress scores between the B1-B3 groups showed no significant difference at all time points of implant placement (*p =* 0.3080). At the 10 s time interval, the values for the B1, B2, and B3 groups were 16.16 ± 35.93, 1.42 ± 3.98, and 2.55 ± 5.37, respectively (Figure 1). At the 30 s time interval, the values for B1, B2, and B3 groups were 26.19 ± 7.28, 8.03 ± 7.77, and 23.82 ± 5.56, respectively. Similarly, at the 57 s time interval, the values for B1, B2, and B3 groups were 22.73 ± 7.6, 11.86 ± 3.55, and 22.73 ± 7.60, respectively (Table 2).

Figure 2 and Figure 3 depict the comparison of the stress scores of bone tap and countersink groups in the cancellous bone at different time points.

### 3.2. Evaluation of Stress Scores in the Cancellous Bone between B1, B2, and B3 Groups at Different Time Points Compared with 10 s as the Time Interval

Stress scores between the B1-B2 groups (*p =* 0.0001) and B2-B3 groups (*p =* 0.0001) were highly statistically significant. Stress scores between the B1-B3 groups showed no statistical significance (*p =* 0.8230). During implant insertion, between the 10 s-to-30 s-time interval, the mean ± (SD) for the B1, B2, and B3 groups were 10.02 ± 33.81, 6.61 ± 7.82, and 21.27 ± 6.81, respectively (Figure 4). Between the 10 s to 57 s-time intervals, the values for the B1, B2, and B3 groups were 6.56 ± 33.28; 10.44 ± 3.83, and 20.18 ± 7.82, respectively (Table 3).

### 3.3. Comparison of Stress Scores in the Cortical Bone between B1, B2, and B3 Groups at Different Time Points

Stress scores between B1-B3 groups (*p =* 0.0001) and B2-B3 groups (*p =* 0.0001) were highly significant at all time points. Stress score comparison between B1-B2 groups only showed a significant difference at 5 s (*p* = 0.0260) and 10 s (*p* = 0.0020). The results for the B1, B2, and B3 groups at the 5 s time point were 134.94 ± 20.68, 90.14 ± 37.88, and 0.00 ± 0.00, respectively. At the 30 s time point, the values for the B1, B2, and B3 groups were 105.99 ± 12.37, 100.88 ± 10.39, and 67.94 ± 12.08, respectively. At the 57 s time point, the resulting values for the B1, B2, and B3 were 125.03 ± 30.67, 114.55 ± 28.63, and 73.16 ± 14.36, respectively (Table 4).

### 3.4. Evaluation of Stress Scores in the Cortical Bone between B1, B2, and B3 Groups at Different Time Points Compared with 5 s as the Time Interval

A comparison between the stress scores in the cortical bone between B1, B2, and B3 groups at different time points with the 5 s time interval showed a significant difference between B1-B2 (*p =* 0.0500), B2-B3 (*p =* 0.0010), and B1-B3 (*p =* 0.0030) groups. The mean ± (SD) for the B1, B2, and B3 groups between 5 s to 30 s time interval during implant insertion were 28.95 ± 18.55, 10.74 ± 40.05, and 67.94 ± 12.08, respectively (Figure 5). The values for the B1, B2, and B3 groups between 5 s to 57 s time intervals were 9.90 ± 31.30, 24.41 ± 45.88, and 73.16 ± 14.36, respectively (Table 5).

## 4. Discussion

The extent of resistance in bone during insertion of an implant is directly proportional to the forces acting on the bony tissues [25]. A stress overload can result in resorption and implant failure. It is vital to reduce stress in every step of implant placement. We used FEM analysis on the model to simulate the stress around an implant and the underlying bone during insertion using three different osteotomy techniques in order to identify which technique resulted in the least amount of stress to the bone. Finite element analysis has been used in previous studies to analyze stress in structures with complicated geometry, such as bone and dental implant systems [15,16]. It provides detailed and accurate qualitative and quantitative results of biomechanical responses.

Increased stress on the bone around the implant inhibits bone formation and leads to the formation of cartilaginous connective tissue [26,27]. The external stresses are transmitted to the osseous tissues through a mechanism known as mechanotransduction [28]. It is the conversion of mechanical energy from stress-induced into bioelectrical and biochemical changes that affect bone cell metabolism. When this energy is excessive, it can lead to cell death of the osteocytes, formation of osteoclasts, and bone resorption. An important factor that influences the long-term success or failure of an implant is the mechanism of stress distribution and transfer to the surrounding bone [13]. In the present study, stress analysis is performed using finite element analysis. There was no consideration of external irrigation during the osteotomy procedure as it was in vitro. However, applying external irrigation (coolant) will help reduce the friction between the drill and osteotomy site, thus helping to reduce the heat generated. Previous studies evaluated the effect of modifying the osteotomy techniques, such as undersizing, bone tap, and countersink, to influence primary stability, bone to implant contact, and osseointegration [29]. We compared the conventional osteotomy method (B1) with the bone tap (B2) and countersink technique (B3), which is believed to reduce the stress on the coronal part of the bone. The stress acting on the cortical and cancellous bone while insertion of the implant was evaluated and compared within the three groups.

We observed higher stresses on the cortical bone than on the cancellous during the insertion of implants in all three groups. In cortical bone, there was a highly statistically significant reduction in the stress scores in the B2 (bone tap) group compared with B1 (conventional) and B3 (countersink). When a pair-wise comparison was made, a significant difference was observed between B1-B2 and B2-B3 groups. These findings concurred with a previous study by Natali et al., who examined who analyzed dental implants in bone and the visco-elastoplastic response of bone tissues [30,31]. They concluded that cortical bone takes up high stress compared to the cancellous and has lesser potential to dissipate stress. Higher torque values could induce higher compressive stresses in the bone around the implant [10]. Increased stress within the cortical bone could lead to damage of crestal bone in cases of high stress.

The bone tap technique is advocated to reduce the overall stress acting on the bone around the implant placement site. Stress primarily occurs at the crestal region of an implant design, where the thread pitch and depth are clinically valuable due to its effect on surface area and speed at which the implant is being inserted [29]. Bone tap helps create a similar thread pitch to the implant being placed, preventing the implant threads from exerting excess stress on the surrounding bone. Thus, the force required for implant insertion is minimized, and the stress distribution is reduced (Figure 1 and Figure 3).

In cortical bone, a significant difference in stress scores was seen between the B1-B2, B2-B3, and B1-B3 groups at time intervals of 5 s to 30 s. B1-B3 (*p =* 0.001) and B2-B3 (*p* = 0.021) groups displayed a statistically significant difference in stress scores from 5 s to 57 s (Figure 4). We observed that B1 (conventional) and B3 (countersink) groups had high bone-implant friction during insertion. The implant threads tend to exert higher force due to friction, necessitating higher energy to guide the implant to its final position. In the B2 (bone tap) group, the threads formed could reduce the frictional resistance. These results are in keeping with previous studies by Aslam et al. [32] and Niroomand et. al. [33], who observed that frictional forces could influence the stress exerted by an implant up to 28.5% in cancellous and cortical bone. The high frictional forces could explain the average stress distribution change from the B1 to B3 and B3 to B2 group.

In cancellous bone, there was a steady increase in stress from 5 s to 45 s. From this point, the stress decreased till the 57 s time point and then was constant. In the B2 group, there was a gradual increase in stress from 5 s to 57 s. The above stress variations are in line with previous studies by Li et al. into mechanical interlocking within the bone and implant and their biological interaction [34].

Within the cancellous bone, the control group displayed the maximum stress (22.73 ± 7.6), followed by B3 (22.73 ± 7.6) and B2 groups (11.86 + 3.55). The results showed a highly statistically significant reduction in stress scores in the bone sink (B2) group compared to the other groups. These results are consistent with data obtained by Steigenga JT et al. [35] and Misch CE et al. [36]. Multiple studies have demonstrated that the thread pitch plays a pivotal role in the stability of an implant in cancellous bone. Smaller pitch implants show better stress distribution. Our results closely tally with those of Williams et al., who demonstrated that in the cortical bone, the maximum stress distribution is located at the area of contact with the implant [37].

In cancellous bone, the B2 group showed a statistically significant reduction in stress scores from time intervals 10 to 30 s and 10 to 57 s. The B2 group displayed an exponential increase in stress scores from the time of implant insertion to completion. The B1 and B3 groups showed a drop in stress scores between 40–45 s and 50–55 s time points in cancellous bone. At 57th second, both B1 and B3 groups showed similar stress scores (Figure 2).

Overall, different osteotomy site preparations may affect the stress distribution. Modifications in osteotomy site preparation may have a significant effect on the exchange of energy. Our study revealed that B2 exhibited significantly lower stress values in the cancellous bone at all time points compared to the B1 and B3 groups. In cortical bone sections, the stress distribution trends for B1 and B2 were constantly fluctuating from 5 s to 57 s. However, for B3, there was a steady increase in stress values in the adjacent cortical bone from 5 s to 45 s. At this point, there was a noticeable decrease in the stress scores. The variations in the stress scores could be due to the difference in the interface conditions, which transfer stress differently. This finding was similar to those obtained by Gumrukcu et al., who compared the bone types in the atrophic maxilla, whereas our study was done on the mandible cut section [38].

These findings may be somewhat limited by the in vitro study design. While FEM analysis offers many advantages, it remains a mathematical model that may not completely replicate the conditions in the oral cavity. Stress analysis is performed under static loading, and the mechanical properties of materials are set as isotropic and linearly elastic, although it may not be the case clinically.

The results of our study suggest that modified osteotomy techniques can be employed to reduce the stress distribution patterns. These findings will help researchers design further studies investigating stress distribution during implant insertion on complete maxillary and mandibular models. Further in vitro research employing cadaveric models, replicating the mechanical and anatomical properties, would confirm and validate these findings.

## 5. Conclusions

This study set out to examine the stresses generated by various osteotomy techniques on bone. Based on our findings, we concluded that the bone tap osteotomy technique exerted less stress on bone compared to the conventional and countersink types of osteotomies. The countersink technique showed lesser stress values in the crestal region. All three groups showed greater stresses on insertion into cortical bone than on the cancellous bone. The bone tap technique resulted in less friction during insertion into cortical and cancellous bone. Overall, the current data suggest that using the bone tap osteotomy technique could reduce the stress on osseous tissues during implant insertion.

## Figures and Tables

**Figure 1 materials-14-07547-f001:**
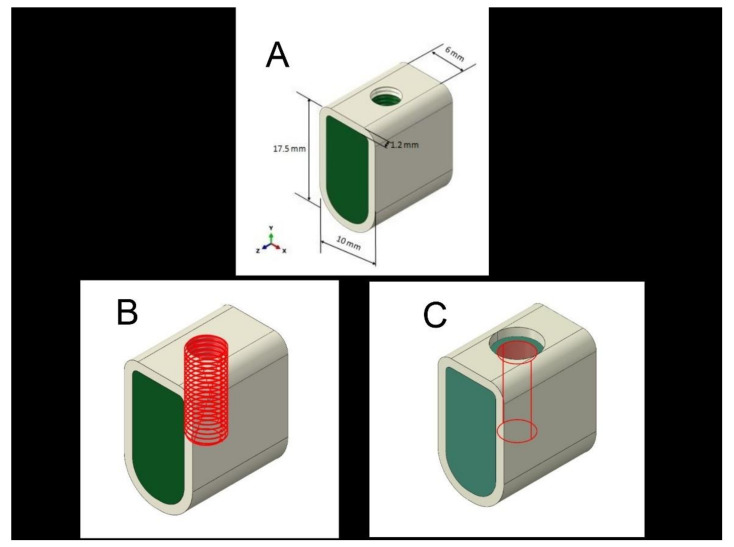
(**A**) Final model for the control group (Conventional osteotomy) (Group: B1); (**B**) final model for bone tap osteotomy (Group: B2); (**C**) final model for Countersink osteotomy (Group: B3).

**Figure 2 materials-14-07547-f002:**
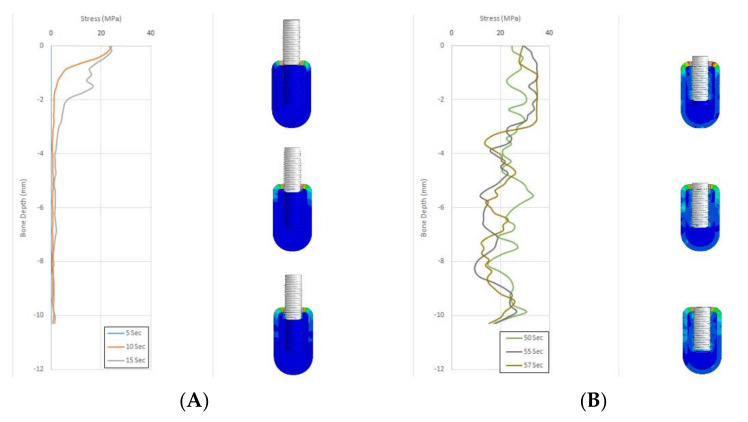
(**A**) Stress scores at 5–15 s for Bone tap osteotomy in cancellous bone. (**B**) Stress scores at 50–57 s for Bone tap osteotomy in cancellous bone.

**Figure 3 materials-14-07547-f003:**
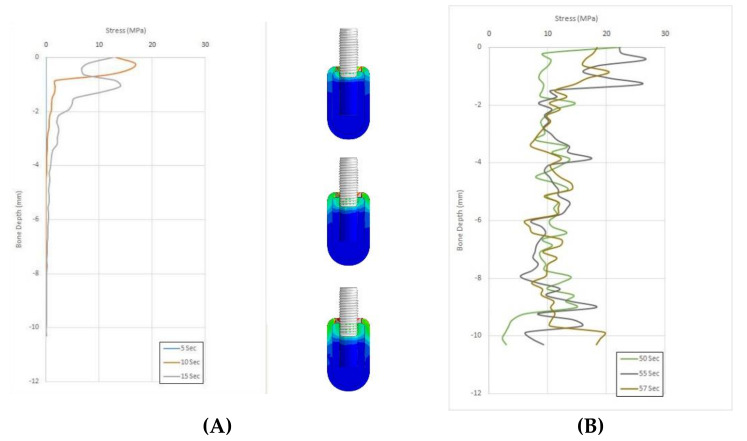
(**A**) Stress scores at 5–15 s for countersink osteotomy in cancellous bone. (**B**) Stress scores at 50–57 s for countersink osteotomy in cancellous bone.

**Figure 4 materials-14-07547-f004:**
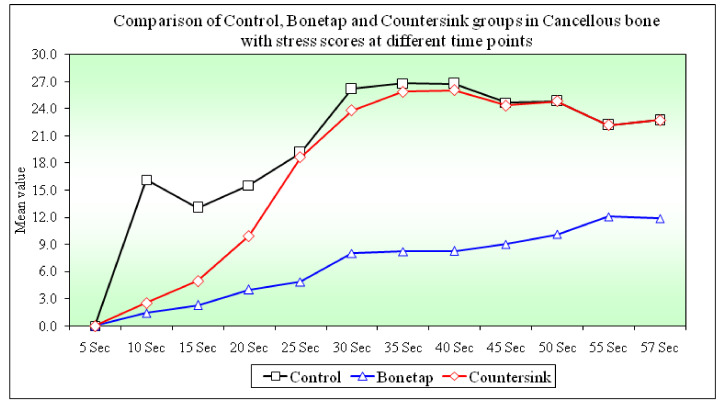
Comparison of Control, Bone tap, and Countersink groups in cancellous bone with stress scores at different time points.

**Figure 5 materials-14-07547-f005:**
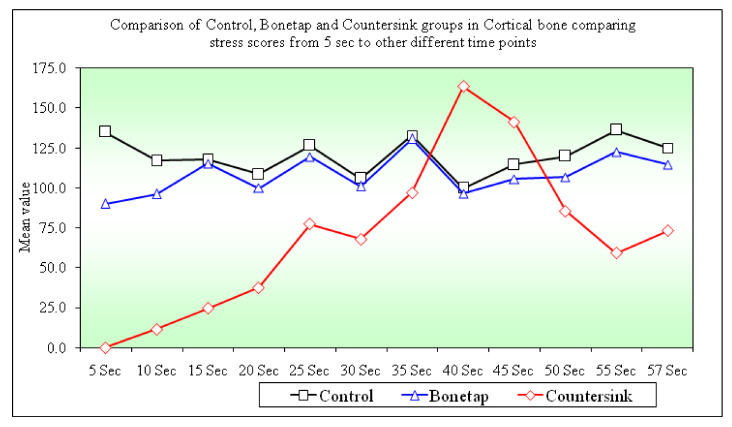
Comparison of Control, Bone tap, and Countersink groups in cortical bone comparing stress scores from 5 s to other different time points.

**Table 1 materials-14-07547-t001:** The physical properties of components in the FEA model [13,25].

	Young’s Modulus (in GPa)	Poisson’s Ratio	Density (in g/cm^3^)	Tensile Yield Strength (in MPa)	Compressive Yield Strength (in MPa)
Cortical bone	15	0.35	1.5	115	182
Cancellous bone	6	0.30	0.67	32.4	51
Titanium implant	113	0.30	4.54	830	830

**Table 2 materials-14-07547-t002:** Comparison of B1, B2, and B3 groups with cancellous stress scores at 10 s, 30 s, and 57 s by Kruskal–Wallis ANOVA followed by Mann–Whitney U test for pair-wise comparisons.

Timepoint (Seconds)	B1	B2	B3	Pair-Wise Comparisons		
				B1 vs. B2	B1 vs. B3	B2 vs. B3
10	16.16 ±35.93	1.42 ± 3.98	2.55 ± 5.37	0.0001 *	0.3080	0.0001 *
30	26.19 ± 7.28	8.03 ± 7.77	23.82 ± 5.56	0.0001 *	0.0700	0.0001 *
57	22.73 ± 7.60	11.86 ± 3.55	22.73 ± 7.60	0.0001 *	1.0000	0.0001 *

* *p* < 0.05 is the significance level.

**Table 3 materials-14-07547-t003:** Comparison of B1, B2, and B3 groups with change scores from 10 s to 15 s, 30 s, and 57 s of time points in cancellous stress scores by Kruskal–Wallis ANOVA followed by Mann–Whitney U test for pair-wise comparisons.

Time Interval (Seconds)	B1	B2	B3	Pair-Wise Comparisons		
				B1 vs. B2	B1 vs. B3	B2 vs. B3
10 to 15	3.10 ± 10.53	0.86 ± 3.44	2.44 ± 3.51	0.4300	0.0020	0.0001 *
10 to 30	10.02 ± 33.81	6.61 ± 7.82	21.27 ± 6.81	0.0001 *	0.8230	0.0001 *
10 to 57	6.56 ± 33.28	10.44 ± 3.83	20.18 ± 7.82	0.0001 *	0.1040	0.0001 *

* *p* < 0.05 is the significance level.

**Table 4 materials-14-07547-t004:** Comparison of B1, B2, and B3 groups with cortical stress scores at different time points by Kruskal–Wallis ANOVA followed by Mann–Whitney U test for pair-wise comparisons.

Timepoint (Seconds)	B1	B2	B3	Pair-Wise Comparisons		
				B1 vs. B2	B1 vs. B3	B2 vs. B3
10	134.94 ± 20.68	90.14 ± 37.88	0.00 ± 0.00	0.0260 *	0.0001 *	0.0001 *
30	105.99 ± 12.37	100.88 ± 10.39	67.94 ± 12.08	0.3680	0.0010 *	0.0010 *
57	125.03 ± 30.67	114.55 ± 28.63	73.16 ± 14.36	0.3680	0.0070 *	0.0040 *

* *p* < 0.05 is the significance level.

**Table 5 materials-14-07547-t005:** Comparison of B1, B2, and B3 groups with change scores from 5 s to 10 s, 30 s, and 57 s of time points in cortical stress scores by Kruskal–Wallis ANOVA followed by Mann–Whitney U test for pair-wise comparisons.

Time Interval (Seconds)	B1	B2	B3	Pair-Wise Comparisons		
				B1 vs. B2	B1 vs. B3	B2 vs. B3
5 to 10	17.89 ± 23.45	6.06 ± 31.61	11.39 ± 7.30	0.1010	0.0340 *	0.9080
5 to 30	28.95 ± 18.55	10.74 ± 40.05	67.94 ± 12.08	0.0500 *	0.0010 *	0.0030 *
5 to 57	9.90 ± 31.30	24.41 ± 45.88	73.16 ± 14.36	0.2660	0.0010 *	0.0210 *

* *p* < 0.05 is the significance level.

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
