# Peer review of "Evaluation of Stress Distribution during Insertion of Tapered Dental Implants in Various Osteotomy Techniques: Three-Dimensional Finite Element Study"

_materials, 2021, doi:10.3390/ma14247547_

Round 1

Reviewer 1 Report

Dear Author

The Ms studied three osteotomy techniques by evaluating the stress exerted by the implant on the bone at different time intervals. The topic is interesting, however the authors ‘guidelines have not been followed.  Moreover, Materials and Methods sections and Results should be significantly improved. For this and other comments I suggest a major revision.

ABSTRACT

The whole abstract should be revised following the authors` guidelines of Materials journal.

Results should be re-written. Avoid repeating mean±SD. Moreover, the values should be written with ± instead of +. For example, 22.73+7.60 into 22.73 ±7.60.

Conclusion should be re-written.

INTRODUCTION

Lines 64-65: Specify the long-term

MATERIALS AND METHODS

Line 127: “ The bone sections were divided into three groups of 5 models based on the different types of osteotomy techniques” then each group was composed by 5 models. Were these models the same? Do they have the same physical properties? Please explain this concept.

Line 135: “Bone cavity prepared with a 1.2 mm deep and 0.5 mm excess 135 diameter preparation than B1 on the crest.” Explain if the excess was on the coronal part, apical part, or the whole preparation.

Line 129: “No tapering was given to any bone cavity.” The Authors should explain this concept.

Line 131: Specify the conventional bone cavity.

I suggest adding a figure of the final cavity achieved in B1,B2 and B3 groups, it should be useful for the readers.

Along the text, mean +SD was repeated, I suggest using it only once time.

Lines 139-145: Please provide a table with all the FEM values used in this study, as young’s modulus, Poisson’s ratio, density, yield stress, and plastic strain values.

Lines 170 : Specify the type of ANOVA used for the statistical analysis.

RESULTS

All tables and figure should be formatted and paginated following the authors ‘guidelines

Line 179-178: I suggest removing “…mean±standard deviation (SD)…” here and along the text

Add figure legends to the tables.

Moreover Figure 1 and 3 might be put together. Moreover, a figure of the calorimetric representation of von Mises stress of B1 should be added.

DISCUSSION

Although the stress created during the drill is correlated to the temperature, during the drill, physiological irrigation is used to decrease this temperature. The authors should discuss this aspect.

Line 257: Provide reference.

Line 263: Provide references about the previous studies.

CONCLUSION

Line 359-360: What did the authors mean by “ex erted”?

Author Response

Reply to the reviewer’s comments

Reviewer 1:

Dear Author

The Ms studied three osteotomy techniques by evaluating the stress exerted by the implant on the bone at different time intervals. The topic is interesting, however the authors ‘guidelines have not been followed.  Moreover, Materials and Methods sections and Results should be significantly improved. For this and other comments I suggest a major revision. 

ABSTRACT

QUESTION 1:

The whole abstract should be revised following the authors` guidelines of Materials journal.

Results should be re-written. Avoid repeating mean ± SD. Moreover, the values should be written with ± instead of +. For example, 22.73+7.60 into 22.73 ±7.60.

Conclusion should be re-written.

ANSWER: Lines 41-56.

Based on the reviewer’s suggestions, the abstract has been reframed. The values have been redefined and the conclusion has been re written. The corrected portion has been highlighted in the manuscript.

INTRODUCTION

QUESTION 2.

Lines 64-65: Specify the long-term

ANSWER:

According to the reviewer’s suggestion, the statement has been redefined. (lines 62-65)

MATERIALS AND METHODS

QUESTION 3:

Line 128: The bone sections were divided into three groups of 5 models based on the different types of osteotomy techniques” then each group was composed by 5 models. Were these models the same? Do they have the same physical properties? Please explain this concept.

ANSWER:

Yes, each group consisted of 5 models, and the models were designed according to the physical properties of bone. It did not change between the models. The variation was only with the type of osteotomy technique. (Line 129-130)

QUESTION 4:

Line 135: “Bone cavity prepared with a 1.2 mm deep and 0.5 mm excess 135 diameter preparation than B1 on the crest.” Explain if the excess was on the coronal part, apical part, or the whole preparation.

ANSWER:

Based on the reviewer’s comments the statement has been modified into “The preparation was done at the crestal region, apically towards the bone cavity” (lines 142-143).

QUESTION 5:

Line 132: “No tapering was given to any bone cavity.” The Authors should explain this concept.

ANSWER:

Based on the reviewer’s the sentence has been modified and re written. A bone cavity with parallel walls was prepared without any tapering. (Lines 132-133)

QUESTION 6:

Line 131: Specify the conventional bone cavity.

ANSWER:

Based on the reviewer’s the sentence has been modified and re written (line 135-137)

QUESTION 7:

I suggest adding a figure of the final cavity achieved in B1,B2 and B3 groups, it should be useful for the readers.

ANSWER:

Based on the reviewer’s suggestions, the figures has been added in the manuscript and has been highlighted. (lines 145-166)

QUESTION 8:

Along the text, mean +SD was repeated, I suggest using it only once time.

ANSWER:

Based on the reviewer’s valuable suggestion, the mean ± SD has been replaced with only the values. The same has been highlighted.(lines 224-226, 274-276).

QUESTION 9:

Lines 139-145: Please provide a table with all the FEM values used in this study, as young’s modulus, Poisson’s ratio, density, yield stress, and plastic strain values.

ANSWER:

Based on the reviewer’s valuable suggestion, the table has been added with all the above values. The same has been highlighted in the manuscript (lines 174-187)

QUESTION 10:

Lines 170 : Specify the type of ANOVA used for the statistical analysis.

ANSWER:

One way ANOVA was used for comparison. The p-value was calculated using the Kruskal-Wallis test ANOVA method. The modified sentence has been highlighted. (lines 212-213)

 RESULTS

QUESTION 11:

All tables and figure should be formatted and paginated following the authors ‘guidelines

The same has been rectified according to reviewer’s suggestions.

Line 179-178: I suggest removing “…mean ± standard deviation (SD)…” here and along the text

Add figure legends to the tables.

ANSWER:

According the reviewer’s suggestions, the rectification has been made.

Moreover Figure 1 and 3 might be put together. Moreover, a figure of the calorimetric representation of von Mises stress of B1 should be added. lines (224-226, 274-276)

DISCUSSION

QUESTION 12:

Although the stress created during the drill is correlated to the temperature, during the drill, physiological irrigation is used to decrease this temperature. The authors should discuss this aspect.

ANSWER:

Based on the reviewer’s suggestion, the sentence has been reframed in the manuscript. Since this was an intro-vitro procedure, we did not use any local irrigant during the osteotomy preparation.

(lines 322-325)

QUESTION 13:

Line 306: Provide reference.

ANSWER:

The references have been added and highlighted in the discussion section. (line 306)

QUESTION 14:

Line 308: Provide references about the previous studies.

ANSWER:

The references has been added and highlighted in the discussion section. (lines 312)

CONCLUSION

QUESTION 15:

Line 359-360: What did the authors mean by “ex erted”?

ANSWER:

The word intended to be exerted and it has been modified and highlighted. (line 410-411)

Reviewer 2 Report

The manuscript entitled “Evaluation of stress distribution during insertion of tapered dental implants in various osteotomy techniques: Three-dimensional finite element study” submitted to Materials aims to evaluate and compare the stress exerted on the bone during implant insertion between conventional and modified osteotomy techniques, performing a dynamic finite element analysis.

This manuscript seems very interesting despite the topic it has already been dealt with in the literature. The study design is complete and the manuscript is written in a quite good English form.

I have some suggestions to improve the quality of the manuscript, enriching the text with further notions.

Introduction

“Implant osteotomy disrupts the bone during contact with the implant”

What do you mean?

“A devitalized bony area of about 1 mm is created around the implant. The bone rejuvenates naturally with a fresh blood supply and indentations within the bone [6].”

Reformulate these sentences or delete them.

“The magnitude of stress generated during the insertion is directly proportional to the bone damage and healing. Lesser stress on the bone allows smaller micromotions between implant and bone and enables optimal implant osseointegration [4]. High stress could result in fibrointegration and lead to an early implant failure[7].”

I am not total in agreement with the authors. It is true that high stress could result in a failure of the implant therapy, however not even. A high insertion torque is required to realize some protocols as immediate loading.

“Literature reveals that the surgical preparation technique influences the forces during implant insertion and impacts the biomechanical, clinical, and biological effects seen after implant insertion”

I suggest to add some recent references after this sentence to reinforce this concept

[doi.org/10.3390/app10238623 - doi.org/10.3390/app11041916]

“Several studies have reported the relationship between the implant geometry and force distribution in a static condition [2,17]”
Please, add more recent literature.

Discussion

“We used FEM analysis to realistically simulate the stress around an implant and the underlying bone during insertion using three different osteotomy techniques to identify which technique resulted in the least amount of stress to the bone”

Please, reformulate this sentence. As realistic as the FEM study may be, it will never be realistic enough to simulate an in vivo model given the lack of bone elasticity and biological factors.

“and countersink technique (B3), which is believed to reduce the stress on the bone”

Please, specify which part of implant site preparation is influenced by the use of countersink (most coronal part)

“This results in a uniform increase of torque.”

Has the implant insertion torque been measured?

“To the best of our knowledge, this is the first study that evaluates the stress dissipated during the implant insertion at different time intervals using different types of osteotomy techniques on cortical and cancellous bone”

Please, reformulate this sentence; there are many studies that evaluate stress dissipated during the implant insertion at different time intervals using different types of osteotomy techniques on cortical and cancellous bone.

Please, add a limitation of this study part at the end of the discussion (non-clinical study, lack of evaluation of the insertion torque, use of synthetic models)

After the corrections suggested by the reviewers I am available for a second round of review.

Author Response

Reply to the reviewer’s comments

Reviewer 2:

The manuscript entitled “Evaluation of stress distribution during insertion of tapered dental implants in various osteotomy techniques: Three-dimensional finite element study” submitted to Materials aims to evaluate and compare the stress exerted on the bone during implant insertion between conventional and modified osteotomy techniques, performing a dynamic finite element analysis.

This manuscript seems very interesting despite the topic it has already been dealt with in the literature. The study design is complete and the manuscript is written in a quite good English form.

I have some suggestions to improve the quality of the manuscript, enriching the text with further notions.

Introduction

 QUESTION 1:

“Implant osteotomy disrupts the bone during contact with the implant”

What do you mean?

ANSWER:

The sentence was not relating to the introduction and hence has been removed from the manuscript. Thank you for your kind suggestion.

 QUESTION 2:

“A devitalized bony area of about 1 mm is created around the implant. The bone rejuvenates naturally with a fresh blood supply and indentations within the bone [6].”

Reformulate these sentences or delete them.

ANSWER:

 The sentence has been reformulated according to the reviewer’s valuable suggestion (lines 84-85).

QUESTION 3:

“The magnitude of stress generated during the insertion is directly proportional to the bone damage and healing. Lesser stress on the bone allows smaller micromotions between implant and bone and enables optimal implant osseointegration [4]. High stress could result in fibrointegration and lead to an early implant failure[7].”

I am not total in agreement with the authors. It is true that high stress could result in a failure of the implant therapy, however not even. A high insertion torque is required to realize some protocols as immediate loading.

ANSWER: 

The sentence has been modified according to reviewer’s suggestions and has been highlighted (lines 90-92).

QUESTION 4:

 “Literature reveals that the surgical preparation technique influences the forces during implant insertion and impacts the biomechanical, clinical, and biological effects seen after implant insertion”

I suggest to add some recent references after this sentence to reinforce this concept

ANSWER:

The references has been added and highlighted according to reviewer’s valuable suggestion (line 101).

QUESTION 5:

“Several studies have reported the relationship between the implant geometry and force distribution in a static condition [2,17]”
Please, add more recent literature.

ANSWER:

The references have been added and highlighted according to reviewer’s valuable suggestion (line 109).

Discussion

QUESTION 6:

 “We used FEM analysis to realistically simulate the stress around an implant and the underlying bone during insertion using three different osteotomy techniques to identify which technique resulted in the least amount of stress to the bone”

Please, reformulate this sentence. As realistic as the FEM study may be, it will never be realistic enough to simulate an in vivo model given the lack of bone elasticity and biological factors.

“and countersink technique (B3), which is believed to reduce the stress on the bone”

Please, specify which part of implant site preparation is influenced by the use of countersink (most coronal part)

ANSWER:

The sentence has been reformulated according to reviewer’s suggestion and has been highlighted (line 329)

QUESTION 7:

“This results in a uniform increase of torque.”

Has the implant insertion torque been measured?

ANSWER: Since this is an in vitro study, the implant torque has not been measured.

 QUESTION 8:

“To the best of our knowledge, this is the first study that evaluates the stress dissipated during the implant insertion at different time intervals using different types of osteotomy techniques on cortical and cancellous bone”

Please, reformulate this sentence; there are many studies that evaluate stress dissipated during the implant insertion at different time intervals using different types of osteotomy techniques on cortical and cancellous bone.

Please, add a limitation of this study part at the end of the discussion (non-clinical study, lack of evaluation of the insertion torque, use of synthetic models)

ANSWER:

The sentence has been reformulated and limitations of the study as quoted by your valuable suggestion has been added (line 395-399).

After the corrections suggested by the reviewers I am available for a second round of review.

Round 2

Reviewer 1 Report

Dear Author

I appreciate the time spent on Ms corrections. However, a few more points should be resolved before the publication.

Figure 1: Please only use the letter to differentiate the images. A,b and c, instead of Figure 1a, Figure 1b and figure 1c. Moreover, why the Figure 1 B have two more sections than Figure 1 A and C? Indeed, a short description of the different sections should be added in the figure legend.

Line 132: Change A in A.

Line 212: Please make the format uniform along the sentences.

Table 1: The authors should add the references of these values if any.

Figure 2: I suggest to the authors to separate the figure 2 in two figures, since the graph would be unreadable. Moreover, the authors are strongly recommended to look other publication in materials and focus on figure legends.

Table 2, 3, 4,5: Please try to make these tables more readable. Moreover, I suggest consulting other publications in Materials.

In the Discussion, delete mean SD as you did in the results section.

Author Response

Reviewer 1:

Dear Author

I appreciate the time spent on Ms corrections. However, a few more points should be resolved before the publication.

Figure 1: Please only use the letter to differentiate the images. A,b and c, instead of Figure 1a, Figure 1b and figure 1c. Moreover, why the Figure 1 B have two more sections than Figure 1 A and C? Indeed, a short description of the different sections should be added in the figure legend.

Ans: The modification has been done based on the reviewer’s suggestions.

Line 132: Change A in A.

Ans: Has been changed

Line 212: Please make the format uniform along the sentences.

Ans: The format has been made uniform.

Table 1: The authors should add the references of these values if any.

Ans: The references have been added and highlighted. (line 166)

Figure 2: I suggest to the authors to separate the figure 2 in two figures, since the graph would be unreadable. Moreover, the authors are strongly recommended to look other publication in materials and focus on figure legends.

Ans: The modifications have been done and highlighted based on the reviewer’s suggestions.

Table 2, 3, 4,5: Please try to make these tables more readable. Moreover, I suggest consulting other publications in Materials.

Ans: According to the suggestion from reviewer, the tables were concised and an attempt was made to make it more readable. The legends and tables were reformatted according to other publications from Materials journal.

In the Discussion, delete mean SD as you did in the results section.

Ans: The mean and SD has been removed from the discussion as suggested by the reviewer. ( Line 356-357)

Reviewer 2:

although the article deals with an inflated topic in the literature, authors improved the quality of the manuscript

Ans: The authors thank the reviewer for the comment

Reviewer 2 Report

although the article deals with an inflated topic in the literature, authors improved the quality of the manuscript

Author Response

Reviewer 1:

Dear Author

I appreciate the time spent on Ms corrections. However, a few more points should be resolved before the publication.

Figure 1: Please only use the letter to differentiate the images. A,b and c, instead of Figure 1a, Figure 1b and figure 1c. Moreover, why the Figure 1 B have two more sections than Figure 1 A and C? Indeed, a short description of the different sections should be added in the figure legend.

Ans: The modification has been done based on the reviewer’s suggestions.

Line 132: Change A in A.

Ans: Has been changed

Line 212: Please make the format uniform along the sentences.

Ans: The format has been made uniform.

Table 1: The authors should add the references of these values if any.

Ans: The references have been added and highlighted. (line 166)

Figure 2: I suggest to the authors to separate the figure 2 in two figures, since the graph would be unreadable. Moreover, the authors are strongly recommended to look other publication in materials and focus on figure legends.

Ans: The modifications have been done and highlighted based on the reviewer’s suggestions.

Table 2, 3, 4,5: Please try to make these tables more readable. Moreover, I suggest consulting other publications in Materials.

Ans: According to the suggestion from reviewer, the tables were concised and an attempt was made to make it more readable. The legends and tables were reformatted according to other publications from Materials journal.

In the Discussion, delete mean SD as you did in the results section.

Ans: The mean and SD has been removed from the discussion as suggested by the reviewer. ( Line 356-357)

Reviewer 2:

although the article deals with an inflated topic in the literature, authors improved the quality of the manuscript

Ans: The authors thank the reviewer for the comment

This manuscript is a resubmission of an earlier submission. The following is a list of the peer review reports and author responses from that submission.